# Peptide Self-Assembled Nanostructures: From Models to Therapeutic Peptides

**DOI:** 10.3390/nano12030466

**Published:** 2022-01-28

**Authors:** Emanuela Gatto, Claudio Toniolo, Mariano Venanzi

**Affiliations:** 1PEPSA-LAB, Department of Chemical Science and Technologies, University of Rome, Tor Vergata, 00133 Rome, Italy; emanuela.gatto@uniroma2.it; 2Department of Chemical Sciences, University of Padua, 35131 Padua, Italy; claudio.toniolo@unipd.it

**Keywords:** atomic force microcopy, hierarchical self-assembly, Langmuir Blodgett peptide films, peptide nanostructures, molecular dynamics simulations, therapeutic peptides

## Abstract

Self-assembly is the most suitable approach to obtaining peptide-based materials on the nano- and mesoscopic scales. Applications span from peptide drugs for personalized therapy to light harvesting and electron conductive media for solar energy production and bioelectronics, respectively. In this study, we will discuss the self-assembly of selected model and bioactive peptides, in particular reviewing our recent work on the formation of peptide architectures of nano- and mesoscopic size in solution and on solid substrates. The hierarchical and cooperative characters of peptide self-assembly will be highlighted, focusing on the structural and dynamical properties of the peptide building blocks and on the nature of the intermolecular interactions driving the aggregation phenomena in a given environment. These results will pave the way for the understanding of the still-debated mechanism of action of an antimicrobial peptide (trichogin GA IV) and the pharmacokinetic properties of a peptide drug (semaglutide) currently in use for the therapy of type-II diabetes.

## 1. Introduction

Peptide-based materials show definite advantages compared to other polymeric materials: (i) they are biocompatible, (ii) their structural and dynamic properties can be tuned by suitable selection of composition and sequence, and (iii) their amino acid components can be easily engineered with the aim to extend the peptide chemistry toolbox beyond the 20 metabolic residues and to endow the peptide scaffold with new functionalities [1,2]. Applications of peptide-based materials span from solar energy conversion and bioelectronics [3,4] to peptide vaccines and anti-cancer therapy [5,6].

Besides these general considerations, there are two fundamental reasons to choose peptide building blocks for the design of supramolecular structures. Chiral selectivity emerges spontaneously from the assembly of such chiral compounds, and, most importantly, peptides are naturally programmed to give rise to hierarchical self-assembly (HSA) [7,8]. HSA begins with the specification of the peptide sequence (primary structure), and it propagates, generating a variety of secondary structures (helices, turns, sheets, etc.) and 3D architectures (coiled-coils, fibrils, tapes, rods, nanotubes, etc.). Non-covalent assembly of these nanostructures produces a superior level of structure that leads to the construction of complex architectures of mesoscopic size [9,10,11,12]. HSA, therefore, represents the mechanism of choice to control the pathway that leads from the single-peptide building block to peptide-based smart materials endowed with the desired function [13,14]. Self-organization, structural complexity, and functionality are strictly related aspects of the same play, i.e., the development, growth, and evolution of bioinspired functional materials [15,16].

Many examples exist in nature of peptides and proteins forming self-assembled nanostructures to exert their functional action: actin molecules elongating as fibrillar structures, microtubules self-assembling for intracellular transport, and collagen triple helices networking to form the extracellular matrix. In bioinspired nanotechnology, peptide materials have been used as nanofibrils (interactions with cells, scaffolds for tissue engineering, etc.), gels (e.g., tissue reconstruction and scaffolds), nanoparticles (drug delivery, bioimaging, biosensing, etc.), and nanotubes (cross-membrane conduits, scaffolds, etc.) [17,18,19,20]. Transition among peptide nanostructures of different architectures and morphologies under physical and chemical stimuli has frequently been observed [21,22,23,24].

The tunability of peptide nanostructures is, at the molecular level, determined by the interplay and delicate balance of electrostatic interactions of polar or charged residues (e.g., Lys, Glu, Asn, and Ser), dispersion forces of nonpolar groups (e.g., Gly, Ala, and Leu), and π–π stacking interactions of aromatic residues (e.g., Phe, Tyr, and Trp). With the aim to enlarge the catalogue of the available building blocks, synthetic peptide chemistry has provided non-metabolic residues, such as C^α^-tetrasubstituted and β- and γ-amino acids [25,26]. Moreover, the synthesis of D-α-amino acids paved the way for the construction of peptide architectures of opposite chirality [27]. Furthermore, peptide-based nanostructures can be endowed with specific functionalities by exploiting the reactivity of some amino acid side chains (e.g., Ser, Cys, Glu, and Lys) for the covalent linking of bioactive groups or conjugation with biocompatible polymers (PEGylation), nucleic acid segments, fatty acids, or glycans [28,29]. It should be emphasized that the progress of peptide chemistry made cost effective the large-scale production and high-standard purification of peptide-based compounds.

There are, however, some severe limitations to the development of peptide drugs. Peptides suffer from instability and after administration, they are usually rapidly degraded by enzymes under physiological conditions [30]. Additionally, bioavailability and unfavorable immune response limit their use in vivo. Peptide self-assembly has also been pursued to obtain peptide-based materials for therapeutic applications [31]. Indeed, nanostructured therapeutic peptides have shown higher stability, improved circulation time, and enhanced targeting capacity and bioavailability, resulting in therapeutic performances superior to those of single-peptide molecules [32]. Besides, the dynamic nature of self-assembled nanomaterials makes easier the reversible disassembling of nanostructures under physiologically sustainable conditions (dissipative self-assembly) [33]. Following this idea, peptide nanostructures of different morphologies have been obtained by tuning the contributions of electrostatic forces, multi-contact van der Waals interactions, directional hydrogen bonds, π–π stacking, and systemic solvophobic effects [34].

In this study, we will highlight some fundamental issues of peptide self-assembly, reviewing our recent work on the formation of peptide structures of nano- and mesoscopic dimensions in solution and on solid substrates. We will initially describe the aggregation properties of some peptide models specifically designed for investigating the role of hydrophobic effects, conformational preferences, and secondary structure ordering to determine the morphology of nano- and mesoscopic structures by HSA. The very same factors also drive the aggregation of bioactive peptides, as we will show when discussing the aggregation properties of an antimicrobial peptide (trichogin GA IV) and a therapeutic lipopeptide (semaglutide). From a fundamental point of view, this review article aims to answer two basic questions: (i) how HSA is affected by secondary structure modifications, and (ii) how the morphology of complex peptide architectures can be tuned by the proper selection of the peptide building blocks and environmental conditions. Molecular formulas and acronyms of the peptide compounds discussed in the following are reported in Figure 1 for clarity.

## 2. The Bottom-Up Approach to Hierarchical Self-Assembly

Within a bottom-up vision, self-assembly can be defined as the spontaneous and reversible association of molecular species to form complex supramolecular architectures according to the physico-chemical information intrinsically contained in the building block components [35,36,37,38]. In this regard, HSA is characterized by the fact that the achievement of a superior level of structure is subject to the complete structuration of the lower level. As far as the design of a molecular device is concerned, functionality and HSA are critically linked by the emergence of new properties at each step of the aggregation process [39]. The control of this spontaneous process is crucial to building a molecular device that is expected to execute with optimal reproducibility and long-time stability billions of operations. The bottom-up strategy to HSA is based on the precise selection of the building block properties in terms of (i) shape, charge, and composition complementarity, (ii) suitable diffusion and mobility, (iii) reversible association, and (iv) sensitivity to environmental changes [40,41].

A dramatic example of HSA is the formation of amyloid fibrils, peptide aggregates detected in degenerative and systemic pathologies (Alzheimer, Parkinson, Creutzfeld–Jakob diseases, and type-II diabetes). In these cases, peptide oligomers organize in β-sheet structures, which lead to the sequential formation of protofilaments, then fibrils, and finally amyloid plaques [42,43,44]. The investigation of the mechanisms leading to amyloid fibrillization has helped in the identification of the short KLVFF motif of the Aβ40 and Aβ42 fragments as the factors responsible for the first steps of peptide aggregation [45,46]. With a reductionist approach, Gazit and co-workers used the FF dipeptide to engineer a variety of nanostructures (tapes, nanotubes, fibrils, etc.) for a number of applications, ranging from sensors to optical and piezoelectric devices [47]. It has been shown that synthetic oligopeptides comprising FF repeats may form nanostructures of different morphologies under proper experimental conditions [48].

Peptide fibrillization was found to proceed through a two-step mechanism, initiated by the formation of nanometric spherulites, generating protofibrillar aggregates, followed by a relatively slow growth of mature fibrillar structures. It has been shown that the latter step is prompted by the formation of extended β-sheet domains [49]. Recently, Knowles et al. introduced a secondary nucleation mechanism, in which toxic oligomers are generated by a secondary nucleation reaction catalyzed by larger fibrils, formed during the first nucleation step [50]. Kinetic analysis showed that the rate of this secondary nucleation reaction depends on both the concentration of monomers and a critical concentration (≈10 nM) of amyloid fibrils. Analysis of the oligomer populations formed during the aggregation of mature Aβ42 fibrils allowed the identification of the elementary steps of the autocatalytic cycle, which include the formation of an heterogeneous population of oligomers of different sizes and structures, the conversion to a β-sheet-rich fibril-forming state, a relative fast dissociation step, and a slower fibril growth [51]. In amyloid peptides, π–π stacking and hydrogen bonding represent directional interactions that guide the self-assembly process toward the formation of amyloid fibrils [52]. X-ray diffraction and solid-state NMR studies highlighted the presence of stacked aromatic groups aligned between β-sheet strands [53] as the structural motif stabilizing amyloid fibers. Extensive all-atom molecular dynamics (MD) simulations in explicit solvent carried out on amphiphilic peptides, characterized by peptide sequences formed by alternate non-polar (A, V, L, F) and charged (K, E) residues, showed that fibril formation depends on the delicate balance of hydrophobic interactions and the propensity to form inter-peptide hydrogen bonds. Electrostatic interactions also contributed to peptide fibrillation favoring the formation of antiparallel β-sheet conformation [54]. In agreement with the secondary nucleation mechanism proposed by Knowles et al. [50,51], MD simulations showed that fibril elongation proceeds through the displacement of vicinal peptides to the surface of the growing β-sheet ladder. The contribution of all-atom MD simulations to provide a detailed picture, on a molecular basis, of peptide self-assembly has been recently reviewed [55,56]. On the experimental side, microscopy techniques with a nanometric resolution also contribute considerably to highlighting the aggregation mechanisms of supramolecular assemblies [57]. For instance, Wang et al. were able to characterize the heterogeneous population of oligomers by high-resolution cryo-electron microscopy experiments displaying different coexisting forms of amyloid fibrils with atomic resolution [58].

Oligopeptides have been finding extensive applications in biomedicine, biomaterial design, and therapeutics [59,60,61]. However, they can also be used to investigate the mechanistic details of aggregation at a molecular level under different experimental conditions (peptide concentration, pH, temperature, ionic strength, etc.). In the following, we will give several examples of model oligopeptides that highlight some fundamental issues of peptide aggregation, specifically (i) the role of sequence modification, secondary structure, and environmental conditions in HSA of peptide foldamers (Section 3.1.1 and Section 3.1.2), and (ii) surface effects on peptide aggregation at the air/water interface and on inorganic substrates (Section 3.1.3). These studies paved the way for analyzing the aggregation propensity of trichogin GA IV, a natural antimicrobial peptide (Section 3.2.1), and semaglutide, a lipopeptide recently commercialized by Novo Nordisk for therapy against type-II diabetes (Section 3.2.2).

## 3. Self-Assembled Peptide Nanostructures: From Models to Therapy

### 3.1. Model Peptides

#### 3.1.1. Aggregation of Helical Peptide Foldamers

Entropy plays a central role in the energetics of self-assembly. The final structure must be reasonably stable at room temperature, but the interactions among the building units should be weak enough for the system to be able to explore a large number of configurations until the configuration of the lowest free energy is achieved. This is because, for complex systems, the most stable structure in a given environment and at a certain temperature originates from a delicate balance between entropy and association enthalpy. A successful strategy to reduce the entropy penalty associated with self-assembly is to use suitably designed, ordered building blocks. In this context, peptide fibrillation is facilitated when the peptide building blocks attain a ß-sheet or a helical coiled-coil conformation [44]. In particular, helical structures may act as key intermediates in the early stage of fibrillation, favoring the formation of small peptide clusters that later evolve into the ß-ladders nucleating the growth of micrometric peptide fibrils [62].

To analyze the role of helical peptide foldamers in HSA, we studied the aggregation behavior of model homo-Aib oligopeptides, where Aib stands for the α-aminoisobutyric acid residue (denoted also by U in a single-letter code notation). The compounds investigated have the general formula Z-(Aib)_n_N, where Z- and N represent the benzyloxycarbonyl- and the –O–CH_2_–CH_2_–(1)naphthyl groups, respectively, (Figure 1a) [63,64], and *n* = 6, 12, and 15. They will be denoted in the following as UnN, with *n* = 6, 12, and 15. As a result of the restriction of the allowed conformational space caused by dimethyl substitution on the C^α^-atom, Aib-rich oligopeptides populate ordered conformations, with a characteristic switch between 3_10_- and α-helical structures with increasing peptide length (*n* > 8) and an increasing number of Aib residues in the sequence (>50%) [65,66,67]. As a consequence, the UnN homo-oligopeptides investigated can be ideally considered as two-turn 3_10_-helix (*n* = 6) and three- (*n* = 12) and four-turn (*n* = 15) α-helix building blocks. The functionalization with an N fluorophore and a benzyl group at the C- and N-termini, respectively, allowed us to analyze how aromatic groups may affect the aggregation process. It was shown that besides enhancing hydrophobic effects, π-stacking interactions among specifically oriented aromatic groups promote the directional growth of peptide fibers [68]. It should be noted that Aib-rich peptides have shown excellent ß-sheet breaker properties, indicating their possible therapeutic use against the formation of amyloid fibril [69].

MD simulations carried out on UnN in methanol/water solutions strengthened the conclusion that in the case of U12N and U15N, the α-helix is the only significantly populated conformer [65]. Moreover, for the longer peptides (*n* ≥ 9) of the series, unfolding events were not registered during the entire simulation time (100 ns). On the contrary, the initial α-helical structure of U6N was lost in a few nanoseconds and frequent switches between 3_10_- and α-helices and, to a minor extent, single ß-turn conformations were repeatedly observed during the MD simulation.

Atomic force microscopy (AFM) experiments performed on UnN deposited on mica from methanol/water solutions showed that in the case of *n* = 6, only globular structures could be observed, as usually found in the case of aggregation of amphiphilic molecules driven by a hydrophobic effect (Figure 1A). However, in the case of U12N (Figure 1B) and U15N (Figure 1C), micrometric filaments were predominantly imaged.

Remarkably, Figure 1B shows micrometric fibers featuring both left- and right-handed helical winding. We are tempted to ascribe this finding to the HSA nature of peptide aggregation, linking the helical morphology of micrometric fibers to the achiral nature of the Aib residue that leads to racemic mixtures of homo-Aib peptides [70]. The helical screw-sense preferences of peptides rich in achiral and chiral C^α^-tetrasubstituted amino acids have been recently reviewed by Toniolo and co-workers [71,72]. It was shown that when a single chiral α-amino acid is inserted at the N-terminus of a homo-Aib peptide, two diastereomeric helices form, with the preference to the helical winding dictated by the stereochemistry of the initial chiral residue, i.e., (L-)α-amino acids favor the formation of right-handed helices. Interestingly, Ceccacci et al. obtained deracemization of a homo-Aib octapeptide using chiral micellar environments. In particular, they induced the formation of a right-handed 3_10_-helix peptide structure in an N-dodecyl(L-)proline surfactant and a left-handed 3_10_-helical conformation for the peptide embedded into the D-enantiomer micelles [73].

These findings clearly illustrate the importance of hydrophobic effects but also highlight the role of stable helical structures in driving the aggregation process. The helical ordering of the peptide chain determines the regular 3D arrangement of the aromatic groups and establishes the nature and extent of the peptide surface accessible to solvent interactions.

Aggeli et al. [74] described accurately how peptides can form through HSA nanostructures of increasing complexity, changing from tapes to ribbons, then to fibers (stacked tapes), and finally to fibrils (entwined fibers). Chiral architectures sprang out from the chiral winding of helical building blocks so that antiparallel β-sheet ribbons showed a left-handed twist, determining, at the upper structural level, helically screwed fibrils [75]. The Aib homo-oligopeptides investigated emphasize two aspects of peptide aggregation: (i) the role of aromatic moieties, i.e., the terminal naphthyl and benzyl groups, and (ii) the rigid helical structure attained by the longer peptides of the series (*n* = 12.15). Figure 1B shows the helical winding of U12N fibers of mesoscopic size, reminiscent of the rigid helical structure attained by the single (nanometric) peptide chain. In the case of U12N and U15N homo-peptides, the regular 3D arrangement of the naphthyl groups gave rise to the formation of fluorescent J-type aggregates, characterized by end-to-end stacked arrays of aromatic groups. The influence of suitably arranged aromatic groups on the morphology of amyloid aggregates has already been investigated [76]. Amyloid formation has been ascribed to a multistep mechanism, the first step of which involves the nucleation of globular aggregates of nanometric size by a hydrophobic collapse [77]. The evolution of these globular clusters to growing elongated fibers is determined by the orientational restriction imposed by an optimal π−π stacking. Other studies have suggested that aromatic−aromatic interactions affect the growth rate of amyloid fibrils [78]. In the homo-Aib systems investigated, the helical conformations stably attained by U12N and U15N promoted the formation of fibril-like structures. MD simulations showed that only in the case of the longer peptides of the Aib series, the regular packing of the peptide chains is made possible, establishing a close connection between the intrinsic stability of the peptide helical building blocks and the capacity to form ordered β-sheet ladders (Figure 2A). In Figure 2B, a sketch of the generation of micrometric peptide fibrils attaining a helical morphology from the twisting of the peptide tapes formed by these β-sheet ladders is also reported.

#### 3.1.2. Peptide Aggregation in Solution: Disrupting Helices

Aggregation is dictated by interfacial phenomena [79]. The stability and morphology of supramolecular architectures result from the balance of enthalpic, i.e., the sum of the interactions among the molecular components, and entropic contributions, that is, the systemic organization of complex structures in a given environment. Recently, it was shown that prefibrillar peptide oligomers, formed by stacks of β-sheet ladders, represent the toxic elements in several neurodegenerative pathologies [80]. Therefore, frustrating the formation of β-sheet conformations would represent a suitable strategy to inhibit the self-assembly process producing amyloid structures. Soto demonstrated that incorporation of β-breaker elements into short peptides able to target amyloidogenic protein definitely hinder amyloid formation [81]. In this regard, Aib features unique β-sheet breaker properties, as can be easily inferred from the analysis of its Ramachandran plot in comparison with those of other β-sheet breaker agents [65]. Interestingly, it was found that the incorporation of an Aib residue into Aβ17–21, a β-amyloid segment detected in Alzheimer disease, induced a conformational transition of the peptide main chain that attained a helical structure in organic solvents [82]. As pentapeptide analogues were shown to form in vitro aggregates that have structural and cytotoxic properties similar to amyloid assemblies, we investigated the aggregation properties of two Ala-based pentapeptides, both functionalized at the N-terminus with a 1-pyrenyl (Py) group, a chromophore exhibiting an intense fluorescence emission in the blue region, that is strongly dependent on the environment polarity. The two pentapeptides (Figure 1b), denoted as PyA5 and PyA3UA, differ by the insertion of an Aib (U) residue at position 4 [83]. The functionalization of the two peptides with a Py group allowed us to analyze the influence of a large aromatic moiety on the aggregation process.

Aggregation of Ala-based oligopeptides has been extensively investigated [84]. It was reported that while A6K forms nanofibers, A9K aggregates as a nanorod, indicating that the length of the hydrophobic tail heavily affects the morphology of the resultant peptide nanostructure. Interestingly, the self-assembly of A6K nanofibers proceeds through the formation of stable intermediate aggregates so that peptide globular structures of different sizes and shapes coexist with short fibers. The formation of A6K nanotubes 20–25 nm in diameter constructed by helically arranged β-sheet ribbons was also observed and structurally characterized by FTIR absorption and solid-state NMR spectroscopy on aligned samples [85].

The study of PyA5 and PyA3UA in MeOH and MeOH/water mixtures allowed us to get some insights into three important aspects characterizing the HSA of peptide nanostructures: (i) the relevance of solvophobic effects, (ii) the influence of the peptide ordering at the secondary structure level, and (iii) the role of π-stacking interactions between aromatic groups. Spectroscopic studies (fluorescence, CD, and IR absorption) consistently indicate that the single Aib vs. Ala substitution dramatically perturbs the conformational landscape and from that, the aggregation properties of PyA3UA and PyA5. In particular, the strong Py/Py exciton coupling observed in the CD spectrum of PyA5 indicates that the peptide main chain attains a helical secondary structure stabilized by short-range π–π interactions independently on the solvent. In contrast, the CD spectrum of the pyrene chromophore in PyA3UA shows a weak chiral signal, as typically occurs for Cotton effects induced by unordered peptide structures.

Consistently with the idea that a stable secondary structure is necessary for fibrillation, AFM experiments on PyA5 films deposited on mica revealed the formation of micrometric fibrils, while in the case of PyA3UA, only globular structures could be imaged. These differences were more evident when the concentration of the deposition solution of the two peptide analogues was increased to 10 mM. Under these conditions, PyA5 formed long, spaghetti-like fibrils (Figure 3A), while PyA3UA self-assembled into micrometric globular structures (Figure 3B). The latter appear as smaller, empty circular (doughnut-like) structures or larger, pancake-like structures characterized by a central knob and rippled borders. These structures are the result of predominant hydrophobic effects with respect to more directional interactions, such as HB and aromatic–aromatic stacking. This is because the rupture of the ordered conformations populated by PyA5 caused by the Aib vs. Ala substitution inhibits the peptide fibrillation, making solvation effects the driving force of the aggregation process. MD simulations carried out for the two peptide analogs starting from five-strand β-sheets formed by helical peptide chains revealed that in the case of (PyA3UA)_5_, the initial ordered arrangement was completely disrupted after a few ns in MeOH, while in MeOH/water solution, amorphous aggregates were rapidly stabilized by a hydrophobic collapse. In contrast, (PyA5)_5_ stacks were found to remain arranged in an orderly manner during the entire simulation. In an aqueous solution, a new peak of the correlation function appears around 10–10.5 Å, indicating the ordered association of two or more sheets. Hydrophobic effects strongly influence the 3D arrangement of peptide aggregates, as can be seen on analyzing the spatial organization of the Py groups. During the MD simulations carried out in methanol/water environments, their separation distances and orientations remained strictly correlated for most of the simulated time in both the pentapeptide analogs, although only PyA5 is able to maintain the initial stacks of β-sheet structures during the 100 ns of the MD simulation.

#### 3.1.3. Peptide Aggregation at the Air/Water Interface

Application of the Langmuir–Blodgett (LB) technique for the construction of interfacial structures with nanometric organization is experiencing an intense revival, finding applications in the design of functional materials in biomedicine, environmental science, and biosensing [86]. In this context, the aggregation of amphiphilic peptides at the air/water (a/w) interface has been investigated for mimicking the toxic interaction of amyloid peptides on the surface of lipid membranes [87,88,89]. In particular, it was found that similar to amyloids, amphiphilic peptides form β-sheet monolayers at the a/w interface [90]. Inspired by our studies in solution, we have investigated the formation of LB peptide films by layering micrometric volumes of PyA5 and PyA3UA chloroform solutions on the water subphase of a LB minitrough [91]. Two important differences characterize the formation of the two peptide films: (i) the onset of the liquid expanded (LE) to liquid condensed (LC) phase transition was observed to take place at much lower molecular mean area (Mma) values for PyA5 than for PyA3UA, and (ii) cyclic LB isotherms of the PyA5 films were found to overlap almost reversibly at each compression/expansion step. On the contrary, in the case of PyA3UA, a remarkable hysteresis of the LB isotherms was observed, suggesting that irreversible changes, most likely due to the formation of heterogeneous 3D-aggregates, take place under compression (Figure 4A).

Interestingly, the fluorescence spectrum of the PyA5 films showed an intense excimer emission, i.e., emission from pyrene–pyrene excited state complexes, indicating the formation of a regular array of stacked pyrene chromophores (Figure 4B). Moreover, when the peptide film was deposited on a solid substrate, the intensity of the pyrene excimer emission steadily increased at each deposition step, suggesting that PyA5 layers stratified on the solid substrate maintaining their 3D ordered arrangement. Such excimer emission is almost negligible in the PyA3UA LB film. These results suggest a completely different structural organization of the aggregates formed by the two peptide analogues at the a/w interface (Figure 4A) and as layered films on quartz substrates (Figure 4B).

AFM imaging of PyA5 films on mica showed peptide fibers of micrometric lengths and nanometric (10–30 nm) thicknesses. At high surface pressures, the densely packed peptide fibers showed remarkable orientational order, dictated by the regular alignment of the peptide fibers under compression (Figure 5C,D).

A deeper insight into the process of the formation and growth of peptide fibers can be envisaged by the AFM images reported in Figure 5A,B, where the peptide fibers seem to spring out from globular structures. Globule-to-fiber transition has often been observed in amyloid peptides [49]. It should be stressed that AFM experiments carried out on PyA3UA LB film produced under the same experimental conditions revealed only the formation of micrometric globular structures, indicating the prevalence of hydrophobic effects. These findings were corroborated by the results of MD simulations carried out for a cluster of 16 PyA5 and PyA3UA building blocks. While the former attained an ordered disposition characterized by a super-helical arrangement of the pyrene groups and regularly aligned peptide chains, mimicking a β-sheet array, the 16 PyA3UA chains collapsed into cluster structures that did not show long-range ordered arrangements either of the aromatic moieties or of the peptide chains. Accordingly, MD simulations produced distance correlation functions g(r) between adjacent PyA5 peptide chains, characterized by a regular ladder of peaks centered at multiple values of 4.8 Å, the typical separation of peptide chains aligned in a ß-sheet arrangement. Such regularity is rapidly lost with increasing distances in the case of PyA3UA. Structural analysis of the PyA5 clusters indicates that aromatic–aromatic and H–bonding interactions concur to stabilize the ordered arrangement of the peptide chains, preparing for the directional growth of peptide fibers.

PyA5 and PyA3UA are a comprehensive case study that illustrates clearly, in our opinion, the significance of HSA. The single-point perturbation of the amino acid composition of the two peptide analogues (the primary level of the structure) strongly affects the conformational landscape of the two peptide chains (the secondary level of the structure). Peptide fibrillation is driven on the sub-nanometer scale by the ordered stacking of the pyrene groups and the ß-sheet-like alignment of the peptide chains. The final morphology of micrometric architectures is dictated by the 3D arrangement of these nanometric structures, i.e., the intertwining of fibrils into peptide fibers (the tertiary level of the structure). The hierarchical nature of peptide self-assembly connects at the different levels the spatial organization of peptide aggregates. In competition with this aggregation pathway, the formation of PyA3UA micrometric globules is the result of the predominance of hydrophobic (systemic) effects with respect to the peptide–peptide interactions that are responsible for the directional growth of peptide nanostructures. This is determined on a molecular scale by the conformational constraint introduced by the Aib residue that inhibited the structuration of the peptide building blocks in a β-sheet array, the precursor of peptide fibrillation. The morphological differences imaged by the AFM measurements between the PyA5 and PyA3UA LB films are similar to that observed in the case of the two peptide analogues aggregates in aqueous solution (Section 3.1.2). In that case, fibrillation was only obtained for PyA5, whereas PyA3UA formed predominantly globular structures. However, it should be stressed that the densely packed arrangement of PyA5 fibers formed at the a/w interface only at the high surface pressures exerted by LB compression. On a molecular scale, the single Aib in place of Ala substitution inhibits PyA3UA to adopt a ß-sheet conformation, hampering the ordered stacking of pyrene groups and the coherent alignment of the peptide chains. Here, we would like to stress that this single-point Aib→Ala mutation affects dramatically the aggregation pathway of the two peptide analogues, irrespective of the environment, i.e., in both aqueous solutions and at the air/water interface (Figure 6). This is a strict consequence of the hierarchical nature of peptide self-assembly, which propagates the structural and dynamical differences of the peptide building blocks to the final morphology of their mesoscopic aggregates.

These results open interesting applicative perspectives. Aib insertion in bioactive peptide sequences could inhibit peptide fibrillation at early stages, representing a promising approach toward the development of a peptide-based therapy against the insurgence of neurodegenerative diseases, as proposed by Gilead and Gazit [69]. On the bionanotechnological side, LB methodology allows the homogeneous coating of solid substrates with nanostructured peptide films, with possible applications in biomineralization, heterogeneous bio-catalysis, antimicrobial coating, and tissue engineering.

### 3.2. Bioactive Peptides

#### 3.2.1. Aggregation of Antimicrobial Peptides

Trichogin GA IV (TrGA) is an antimicrobial peptide of the peptaibol family [92,93], having the sequence nOct-Aib-Gly-Leu-Aib-Gly-Gly-Leu-Aib-Gly-Ile-Lol (nOct- = n-octanoyl group; Lol = 1,2-aminoalcohol leucenol). TrGA predominantly adopts a mixed 3_10_-/α-helix structure, specifically a distorted 3_10_-helix at the N-terminus and a longer α-helical segment at the C-terminus. The TrGA structure is characterized by a hydrophobic region formed by the n-octanoyl group and the Leu and Ile side chains. The four Gly residues are positioned in the opposite surface, with the Aib residues being aligned on the borderland between the two surfaces. The structural and dynamical properties of TrGA were thoroughly analyzed by us applying time-resolved optical spectroscopy techniques and theoretical conformational analysis [94,95]. These studies revealed that a conformational transition between a helical structure and a bent conformation characterized by a turn around the central Gly-Gly residues was taking place in the microsecond region. It was also shown that such equilibrium can be shifted toward the latter conformation by the association of metal ions, such as Ca(II), Tb(III), and Gd(III) [96]. We also used a thiolated TrGA to anchor lipid bilayers into polymeric nanocavities arrayed on a gold platform [97].

Recently, TrGA films formed at the a/w interface under LB compression were characterized by MD simulations and AFM imaging [98]. At low surface pressures (LE phase), the peptide chains lay almost parallel to the water subphase, adopting preferentially a helical conformation. In the LC phase, TrGA formed globular aggregates that, on increasing the surface pressure, gave rise to fibrillation. At this stage, the peptide fibers organized as networks of meshes confining small water pools. At higher surface pressures, TrGA collapsed as 3D aggregates (solid phase), in which the peptide chains attained a stretched conformation and aligned vertically with respect to the water subphase. To analyze the influence of the dynamical properties of TrGA on the processes that lead to the formation of the LB films, we synthesized a conformationally constrained TrGA analogue, characterized by the substitution of the Gly residues at positions 2, 5, and 16 with hydrophobic Leu residues. The Gly residue at position 9 was also suppressed [99]. These changes severely restrain the conformational region accessible to the trichogin analogue, denoted in the following as TrGAr. Theoretical conformational analysis and spectroscopic results demonstrated that TrGAr adopts a rigid helical structure under the applied experimental conditions.

LB experiments revealed remarkable differences in the behavior of the two peptide analogues investigated. While, during the LB compression of the TrGA film, the surface pressure increased monotonically until the collapse of the LB isotherm (Figure 7A), the TrGAr isotherm showed the typical trend of a first-order transition, characterized by a constant surface pressure interval in which the LE and LC phases coexist (Figure 7C). Accordingly, the compressibility modulus (K_M_) of TrGA steadily increases, until the collapse of the LB film at high surface pressures (solid phase), when a sudden decrease of K_M_ can be observed (Figure 7B). In the case of TrGAr, almost null K_M_ values are associated with the flat region corresponding to the LE/LC coexistence (Figure 7D), indicating the occurrence of a quasi-reversible transition characterized by a marked reduction of Mma without effect on K_M_. These findings are in fair agreement with MD simulations that showed that, at high surface pressures, the helical content of TrGA markedly decreases in favor of turn/coil conformations, while TrGAr maintains a predominant helical structure under the applied surface pressure conditions.

AFM imaging of TrGA films supported on hydrophobic highly oriented pyrolytic graphite (HOPG) or hydrophilic (mica) surfaces showed micrometric globular structures on both substrates, although in the hydrophobic substrate, some incoming fiber network may appear (Figure 8A,B). On the contrary, LB films of TrGAr on mica (Figure 8D), but not on the hydrophobic HOPG (Figure 8C), showed a dense layer of peptide rods of nanometric thickness and micrometric length.

These results demonstrate that the morphology of peptide films, coating homogeneously macroscopic regions of the water subphase, is determined through HSA by the conformational and dynamic properties of the peptide chains and these features also affect the entire self-assembly process at the different levels of organization. The Leu vs. Gly substitutions that differentiate TrGAr with respect to TrGA gave rise to three important effects that deeply affect the different steps of HSA: (i) on a molecular scale, the restriction of the conformational landscape of the rigid analogue, (ii) on the nanometric scale, the different structuration of the peptide oligomers seeding the formation of peptide aggregates, and (iii) on the mesoscopic scale, the morphology of peptide films coating extended regions of the solid substrates. This is simply because the rigid helical conformation characterizing TrGAr in solution was also maintained in the peptide film, independently on the applied surface pressure. On the contrary, in the case of TrGA, systemic solvophobic effects predominate on the directional interactions, which could promote the growth of peptide fibers. As a consequence, independently of the nature of the substrate and of the applied surface pressure, only globular structures could be imaged by AFM experiments.

#### 3.2.2. Aggregation of Therapeutic Peptides

Improvements in the pharmacokinetic profile of therapeutic peptides have been achieved through several strategies, among others, the incorporation of non-coded amino acids and the derivatization or conjugation with fatty acids [100] or cholesterol [101]. Longer retention times of lipopeptides have been obtained limiting renal secretion, promoting either the formation of peptide nanostructures or the association to blood proteins, in particular human serum albumin [102].

Recently, we investigated the aggregation properties of semaglutide (SMG), a therapeutic 37-mer peptide against type-II diabetes [103]. In the SMG analogue investigated, the Ala residue at position 8 was replaced with an Aib, and the Lys at position 26 was derivatized in the side chain by two consecutive PEG groups, a γ-Glu residue. Furthermore, a C_18_–OH lipid chain was covalently linked to the Glu side chain to increase the SMG affinity to human serum albumin. The insertion of an Aib residue at the N-terminus would contribute to inhibit enzymatic degradation. Early on, SMG showed good solubility in aqueous solutions, as only monomers and small oligomers (mainly dimers) could be devised from the small hydrodynamic volumes and the fast correlation times predicted by fluorescence anisotropy measurements and MD simulations, respectively [103]. In contrast, at long lengths of time (weeks), experimental evidence strongly suggested that a steady aggregation process was taking place. SMG self-assembly led to the formation of peptide aggregates, characterized on the molecular scale by peptide chains aligned in a ß-sheet conformation and, on the micrometric scale, by extended dendrimer-like morphologies coating homogeneously the mica substrate. The observed cooperativity of the process is clearly the result of HSA, nucleated by the peptide oligomers formed earlier.

MD simulations showed the quite compact structure attained by the SMG monomer (Figure 9A), dimer (Figure 9B), and trimer (Figure 9C). The charged side chains of the SMG oligomers point out to the solvent, while the C18–OH lipid chain fold, protecting the hydrophobic core of the protein. Aromatic–aromatic interactions involving the Trp and Tyr residues contribute to stabilizing the compact structure of peptide oligomers. The observed hypochromic effect of the SMG UV absorption spectrum and the fluorescence emission quenching strengthened these conclusions.

The kinetics of SMG aggregation are characterized by a time-dependent rate constant, typical of a fractal autocatalytic process, in which nucleation is the rate-determining step. The lag time for the onset of SMG aggregation is associated to a critical concentration of peptide clusters that triggers the fast autocatalytic growth of large aggregates.

Interestingly, the catalytic rate constant of the process increases proportionally to the size of the aggregate, s(*t*), which in turn depends on time through a characteristic power law, i.e., s(*t*)^n^ [104]. Within this model, SMG aggregation occurs through a multistep mechanism that requires (i) the fast formation of peptide oligomers (mainly dimers), as suggested by MD simulations and fluorescence anisotropy measurements, followed by (ii) a random lag time, generally long-lasting, that ends when a critical concentration of peptide clusters is achieved, and finally (iii) a fast and highly cooperative step associated with the formation of micrometric aggregates. Interestingly, CD spectra showed that large-scale aggregation is associated with an abrupt change in the SMG conformational landscape, from populating random coil and helical conformations to a predominant β-sheet structure.

AFM imaging of micromolar SMG solutions deposited on mica immediately after being prepared revealed the formation of nanometric globular structures. In contrast, aged SMG solutions were shown to form, under the same experimental conditions, dendrimer-like structures built by the self-assembly of nanometric peptide rods (Figure 10A). These highly organized structures appeared after considerably long time periods, following the cooperative burst of aggregation. The observation of these fractal structures strongly supports the multistep mechanism described above and used for reproducing the aggregation kinetics reported in Figure 10B.

The study of the aggregation potential of therapeutic candidates is of fundamental importance for the design of peptide drugs featuring optimal pharmacokinetics. The case of semaglutide, a drug already in the market, where binding to human serum albumin is pursued as a half-life extending strategy and fibrillation is frustrated as part of the drug optimization process is paradigmatic [105].

## 4. Conclusions

Peptide self-assembly is a pervasive process in nature that produces a large collection of structures on the nano- and mesoscopic scales. The morphology of these supramolecular architectures (nanotubes, nanotapes, fibers, coiled coils, etc.) can be controlled by selecting the proper environmental conditions and by a fine balance of the interactions governing the aggregation of peptide building blocks. In this study, we have shown that peptide fibrils are formed when the peptide chains form sufficiently long (and stable) helices [63,64]. In this case, the network of hydrogen bonds represents a directional constraint for the growing aggregate. This effect can also be achieved through a regular array of aromatic groups stabilized by π–π interactions [83]. On the contrary, globular structures have invariably been found where hydrophobic effects predominate, causing the hydrophobic collapse of the nascent nanostructure.

The principal feature of peptide self-assembly is its hierarchical nature, which propagates the secondary structure properties of the peptide building block through the different levels of HSA until the achievement of the final morphology of micrometric structures. The observation that a single amino acid substitution is able to inhibit the formation of amyloid fibrils paves the way for the design of new-concept therapeutic peptides fighting neurodegenerative diseases [83]. This effect was verified also in the formation of Langmuir–Blodgett films of model [91] and antimicrobial [99] peptides, where we demonstrated that the homogeneous coating of large areas of solid substrates is facilitated when fully developed peptide helices are used. The strategy behind the case studies discussed in this contribution is based on the restriction of the conformational landscape of the peptide building blocks with the aim to reduce the entropy penalty associated with the formation of aggregates in solution or thin films at the air/liquid interface. Lipidation is an alternative way to control the structuration of peptide aggregates, affecting the pharmacokinetics and bioavailability of therapeutic peptides [103]. The still-open question is if peptide nanostructures may affect per se the target/drug recognition process or if they simply impinge upon the secretion processes determining the drug retention.

These results may unfold encouraging perspectives for biomedicine (formulation of therapeutic peptides, design of inhibitors of fibrillation and amyloid genesis, scaffold for regenerative medicine and tissue repair, etc.) and bionanotechnology (antifouling coating, biomineralization, bioinspired nano- and microdevices, etc.) [106].

## Data Availability

Not applicable.

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
