# Peer review of "Peptide Self-Assembled Nanostructures: From Models to Therapeutic Peptides"

_nanomaterials, 2022, doi:10.3390/nano12030466_

Round 1
Reviewer 1 Report
Dear editor,
In this manuscript, the authors provide a review of the literature related to peptide self-assembly. The emphasis of the review is on alpha-helical peptides for therapeutic uses (e.g., anti-bacterial peptides). The authors have worked on this topic and they provide an interesting perspective in the manuscript, which is well written. However, three important developments in the field of peptide self-assembly are missing from the review and they need to be discussed for the manuscript to be of interest to the community. Thus, before I can recommend the manuscript for publication, the authors need to discuss:
1) In page 3 (line 122), the authors discuss the two-step mechanism of fibril formation. This was the most accepted mechanism of fibril formation five years ago but not anymore. Now, it is commonly accepted that fibrils form following a secondary-nucleation mechanisms. The authors should write one or two sentences about it (even if they disagree with it). Please cite:
b) The ability of all-atom simulations to describe the spontaneous formation of amyloid fibrils. This first such studies was published recently and it was made possible thanks to faster computers and better models (i.e., force fields). Please cite:
https://doi.org/10.1016/j.molliq.2021.118283
3) While the manuscript is well written, many/most of the paragraphs in the manuscript are made of just one sentence. Paragraphs are usually made of three to five sentences. This can be easily fixed and it will help organize ideas in the manuscript in a logical manner.
Reviewer 2 Report
The manuscript by Emanuela Gatto described that recent progress of nanomaterials formed by hierarchical self-assembly of designed peptides including therapeutic peptides. I think the manuscript is well written. I recommend that this manuscript is published after minor revision as follows.
- The authors should cite recent reviews more as example of bioinspired peptide nanomaterials. For example, Chem. Soc. Rev. 2015, 44, 8288; Chem. Soc. Rev., 2014, 43, 2743; Chem. Commun., 2018, 54, 8944; Bull. Chem. Soc. Jpn. 2019, 92, 391; Bioactive Materials 2022, 11, 268; Chem. Soc. Rev. 2018, 47, 3659; Adv. Sci. 2019, 6, 1802043; Nano Today 2017, 14, 16.
- Figure 1(B,C) shows the formation of helical assemblies from U12N and U15N. The authors should mention the helix sense (right hand, left hand, or their mixture) and the pitch of the assemblies. Is it possible that the sense and pitch are controlled by the self-assembling conditions?
- The micrometric donut-like structures self-assembled from PyA3UA (Figure 3B) is very interesting structure. The authors should explain the formation mechanism of such structures enough. Can the MD simulation explain the formation of donut-like structures?
Author Response
Please, see the attachment

Reviewer 3 Report
Mariano Venanzi et al. gave a well-organized review on peptide self-assembled nanostructures. It is updating and of vital importance in the field of peptide-based supramolecular assembly. I think it attracts great attention and will have a wide readership. Basically, I would like to recommend it to be accepted by this journal after considering the following points.
- If possible, the authors provide chemical structures of modified and unnatural peptides.
- Also, some important schematic illustrations should be added to improve the quality of this review.
- The letters in all Figures should be reorganized and uniformed.
- It will be much better if the authors give morerelevant applications of peptide self-assembled biomaterials, such as energy conversion, cancer therapy.
- Some updating reports are recommended to add in the part of References, such as CCS Chemistry, 2021, 3, 8-16; Angew. Chem. Int. Ed., 2020, 59, 18960-18963; Chem. Int. Ed. 2019, 58, 11072-11077.
Author Response
Please, see attachment

Round 2
Reviewer 1 Report
Dear editor,
The authors have addressed all my concerns appropriately. I therefore recommend the manuscript for publication as it is.